# Deleterious Alteration of Glia in the Brain of Alzheimer’s Disease

**DOI:** 10.3390/ijms21186676

**Published:** 2020-09-12

**Authors:** Eunyoung Kim, Undarmaa Otgontenger, Ariunzaya Jamsranjav, Sang Seong Kim

**Affiliations:** College of Pharmacy, Hanyang University ERICA, Gyeonggi-do 15588, Korea; ukino0608@gmail.com (E.K.); undarmaa@hanyang.ac.kr (U.O.); ariunzaya0725mnums@gmail.com (A.J.)

**Keywords:** Alzheimer’s disease (AD), glia, microglia, astrocyte, neuroinflammation, cytokine, microRNA, mGluR, complementary system, exosome, glutamate, excitotoxicity, ion channel, synapse

## Abstract

The deterioration of neurons in Alzheimer’s disease (AD) arises from genetic, immunologic, and cellular factors inside the cortex. The traditional consensus of the amyloid-beta (Aβ) paradigm as a singular cause of AD has been under revision, with the accumulation of exploding neurobiological evidence. Among the multifaceted casualties of AD, the involvement of glia gains significance for its dynamic contribution to neurons, either in a neuroprotective or neurotoxic fashion. Basically, microglia and astrocytes contribute to neuronal sustainability by releasing neuroprotective cytokines, maintaining an adequate amount of glutamate in the synapse, and pruning excessive synaptic terminals. Such beneficial effects divert to the other detrimental cascade in chronic neuroinflammatory conditions. In this change, there are new discoveries of specific cytokines, microRNAs, and complementary factors. Previously unknown mechanisms of ion channels such as Kv1.3, Kir2.1, and HCN are also elucidated in the activation of microglia. The activation of glia is responsible for the excitotoxicity through the overflow of glutamate transmitter via mGluRs expressed on the membrane, which can lead to synaptic malfunction and engulfment. The communication between microglia and astrocytes is mediated through exosomes as well as cytokines, where numerous pieces of genetic information are transferred in the form of microRNAs. The new findings tell us that the neuronal environment in the AD condition is a far more complicated and dynamically interacting space. The identification of each molecule in the milieu and cellular communication would contribute to a better understanding of AD in the neurobiological perspective, consequently suggesting a possible therapeutic clue.

## 1. Introduction

As the lifespan of the modern population continues to increase, dementia becomes a central concern in public health incurring a bewildering amount of spending for the treatment. The recognized traits of dementia encompass malfunction of memory formation and recall, rational reasoning, and stereotaxic movements, commonly found in senile people. Among various outcomes of dementia, Alzheimer’s disease (AD) is the most common type, comprising more than 50% of cases [1]. The cause of AD can be either hereditary or sporadic occurrence by a myriad of corresponding genes and environmental factors [2]. Regardless of the origin of onset, the consequence results in neuronal death due to pathogenic protein plaque accumulation, namely, amyloid-beta (Aβ). The generation of Aβ_1–42_ trimmed from amyloid precursor protein (APP) by β-secretase produces plaque formation in the cortex precipitating on neurons and glia [3]. Intracellularly, neurofibrillary tau protein is entangled through the breakdown of microtubules, eventually leading to the destruction of neurons [4]. External as well as internal plaque accumulations are considered primary causes of AD development so that most of the therapeutic strategies have been focused on removing plaque accumulation. After decades of research, however, most AD therapy candidates have failed to revert the symptom, even though they can faithfully remove Aβ plaque accumulation in the cortex [5]. This seemingly disappointing result recalls a new paradigm of the concept in regard to the main cause of AD. Recently, many research groups have proposed a more sophisticated interplay of inflammatory and immunologic agents in conjunction with astrocytes and microglia (Figure 1). New evidence reveals that Aβ functions as just one of the causal factors over the course of a cascade of events in neuronal deterioration.

In this review, we discuss the latest progress in the study of neuroinflammatory interactions in AD progression. Memory impairment as the most outstanding symptom of AD can be understood in the aspect of synaptic loss, which accounts for the pathological turnover of glia and its following engulfment of synapses. In glial activation, various cytokines, complements, microRNAs, receptors, and ion channels are synergistically involved in the cortical milieu, influencing detrimental effects on neurons.

## 2. Dynamic Display of Microglia Polarization in Neurodegeneration

### Maintenance of Neuroprotective Effect of Microglia in Neurodevelopment and Mild Inflammatory Condition

As a surveilling agent in the central nervous system (CNS), microglia stand in a central part of the CNS immune system. In cooperation with the complementary system, it engulfs pathologic protein entanglements, as well as damaged cellular components, and organic toxins to maintain immunological homeostasis. During the procedure, it can sense exogenous particles by specific receptors expressed on the membrane surface such as scavenger receptors (SR-AI/II), CD36, Fc receptors, RAGE (receptor for advanced glycosylation end products), TLRs (toll-like receptors), TREM2, and complement receptors [6,7,8,9]. Particularly in the AD case, it can eliminate Aβ plaques through phagocytosis with apoE lipoprotein at the early stage of disease progression. TREM2, a membrane receptor expressed in microglia, can recognize apoE encircling Aβ plaques leading to phagocytosis [10,11]. Likewise, the chemokine receptor, CX3CR1, in microglia could modulate its phenotype to produce a microglial barrier by opsonizing Aβ plaques, resulting in a less neurotoxic Aβ form [12].

With the phagocytic role, it can act as a key player in neural development through synaptic pruning [13]. Microglia secrete several synaptogenic factors such as brain-derived neurotrophic factor (BDNF), interleukin 10 (IL-10), and interleukin 1β (IL-1β) [14,15]. In the early neural development stage, an excess of dendritic spines and immature synapses builds unorganized brain circuitry [16]. To construct efficient synaptic circuitry, microglia eliminate such redundant synapses via crosstalk with other glial cells [16,17].

Microglial synaptic pruning has been mostly explained in terms of complement proteins such as C1q and CR3 [18]. Recently, complementary-independent synaptic pruning of microglia has also been suggested with the result that CR3 deletion still represents the synaptic pruning effect [19]. Some current studies show that microglia could selectively engulf supernumerary synapses in response to phosphatidylserine (PS), so-called “eat me” signals released from neurons. On the contrary, it bypasses engulfment when transmembrane immunoglobulin CD47 is expressed in neurons as a “don’t eat me‘ signal [20,21].

## 3. Complement Pathway of Microglia in the Detrimental Effect during AD Progression

As AD progresses, however, those beneficial effects turn into adverse outcomes when the microglia become overstimulated by increasing harmful neuroinflammatory factors. Now, it is recognized for the adverse roles in AD progression such as secretion of neurotoxic factors, synapse engulfment, and even tau pathology exacerbation [22,23,24,25,26]. Recently, many studies indicate the activated microglia as a culprit of neuronal synaptic loss, causing long-term potentiation (LTP) impairment and neural circuit disintegration [27]. In normal brain development, the classical complement pathway between neurons and microglia plays a crucial role in constructing local neuronal circuitry through the synapse pruning process [13]. While C1q and C3 are highly expressed at the axon terminal, complement receptor 3 (CR3) in the microglial surface recognizes them to engulf the synapses by phagocytosis [28,29]. However, in AD progression, stimulated microglia produce an excessive amount of C1q, then self- activatethrough CR3 interaction [30]. Thus, microglia initiate hyper-neuroinflammation, resulting in neuronal damage deepening the disease pathology [31]. In a toxic environment, inflammatory elements like lipopolysaccharide (LPS), Aβ oligomers, and tau entanglement can even more activate microglia through the complement cascade [32]. For example, the activated microglia in presence of inflammatory components (Aβ, LPS, and tau) release a sialidase, the enzyme removing a sialic acid residue in the terminal region of sugar chains of glycoproteins or glycolipids expressed in the microglia cell surface. Once desialylated, CR3-mediated phagocytic activity of microglia becomes enhanced [32]. Likewise, microglia install various mechanisms of self-stimulation, actively destroying synapses.

Meanwhile, activated microglia fall into two different polarized states, namely the classical M1 and the alternative M2 phenotype, with almost opposite functions [33]. The M1 phenotype mediates neuroinflammatory cascade, while M2 exerts neuroprotective enrollment. The induction to M1 phenotype is initiated through exaggerated immune stimulation by neuroinflammatory mediators, lipopolysaccharides (LPSs), and interferon γ (INF-γ), secreting pro-inflammatory factors like nitric oxide (NO), TNF-α, IL-6, and IL-1β [33,34]. On the contrary, other types of interleukins such as IL-4 and IL-13 transform microglia into M2-type releasing neuroprotective cytokines such as IL-10, IL-33, TGF-β, and IGF-1 [35,36,37,38]. Given the characteristic of transitional activation under a different impetus, microglia have the freedom to stand on either side of neurological consequences depending on the environmental condition. Once the balance is weighted on M1 conversion, it cascades into further neuronal deterioration in the presence of tau proteins, not to mention external Aβ oligomers.

Like the pathological conversion of Aβ, hyperphosphorylated tau oligomer causes microtubule collapse then forms neurofibrillary tangle (NFT), leading to brain atrophy and cognitive decline [39]. Previous studies have demonstrated that microglia instigates tau pathology in diverse ways, inducing tau aggregation by proinflammatory cytokine release [40,41,42] and spreading hyperphosphorylated tau oligomers or NFT through exosome secretion [43,44]. Interestingly, a lot of evidence addresses the strong correlation between the complement pathway and tau pathology. In the human AD brain and tauopathy mouse model, C1q levels were positively correlated with tau aggregation, and C3-C3aR1 expressions were increased proportional to the progression of tau pathology harming the cognitive function [45,46]. Since microglia likely phagocytose C1q-tagged synapses leading to synapse loss, tau entanglements induce accumulation of C1q at the synapse [45]. Once the complement cascade is initiated by the activation of C1q, C3-C3aR1 signaling can exacerbate tau pathology through the signal transducer and activator of transcription 3 (STAT3) pathway inducing neuroinflammation [46]. Thus, inhibition of the complement pathway can be suggested for reduction in tauopathy. For example, C5a receptors (C5aR) were closely involved in NFTs in human brains and C5aR antagonists decrease tau pathology in the AD mouse model [47,48]. An increase in membrane-attack complex (MAC) formation led to increased tau pathology and neuronal loss [49]. Overexpression of C3 inhibitor protein sCrry or deletion of C3aR were also found to decrease tauopathy [46,49]. On the other hand, other studies have illustrated that microglia contribute to elimination of tau deposits by phagocytosis [50,51,52]. The role of the CX3CR1 signaling pathway of microglia has been demonstrated as a key factor to phagocytose tau entanglements [53]. In normal neuronal development, cytokine CX3CL1 (fractalkine) expressed by neurons binds to CX3CR1 of microglia maintaining resting state, inhibiting pro-inflammatory cytokine release [54]. In the AD condition, on the other hand, tau directly binds to CX3CR1 provoking microglia, which is related to tau internalization by microglia [53].

In the most recent studies, two different groups demonstrated that microglia are not the agitators of tau aggregation. They suggest, however, different results when it comes to tau deposits and its clearance dependent on microglial activation. Zhu et al. show the involvement of microglia in tau pathology over the disease progression using hTau mice in normal conditions of microglia [55]. Interestingly, microglia remained in homeostatic status throughout the whole life span of hTau mice without any disease-associated gene expression or functional phenotype change of microglia. Even the reduction in the number of microglia did not increase tau entanglements, questioning the active involvement of microglia in reducing tau pathology exacerbation [55]. In the other study, Lynn et al. used Thy1-hTau.P301S (PS) mice to evaluate the interaction between tau aggregation and microglial activation [56]. They showed that pTau aggregation predated activation of microglia in the cortex, which provoked microglial phenotype change such as morphological dystrophy, homeostatic marker loss, and lysosomal swelling. With lysosomal swelling of microglia, pTau oligomers and postsynaptic structures like dendritic spines were observed in the microglial lysosome, proving that activated microglia truly phagocytosed the pathological form of tau oligomers as well as engulfed synapses. They concluded that the microglia were activated by pTau aggregation and conducted the traditional surveilling role in AD progression [56]. Thus, the functional characterization of microglial activation with phosphorylated tau oligomer still remains elusive.

## 4. The Physiological Perspective of Microglial Activation

Several ion channels are involved in a functional change of microglia in physiological and pathological conditions. Even though microglia are considered as non-excitatory cells, manipulating neuronal excitability via control of discrete ion channels and their related proteins could affect AD pathogenesis [57,58]. In addition, expressions of ion channels in microglia are notably dependent on the functional state of cells [59]. For instance, the LPS-caused neurotoxic conversion to M1 microglia induces high current densities of Kv1.3 with no current of Kir2.1 in the presence of IL-4, while neuroprotective M2 polarization leads to the immense current of Kir2.1 with no Kv1.3 current [60]. Besides, various research has shed light on ion channels (e.g., Kv1.3, Kir2.1, SK3, KCNN3, KCNN4, KCa3.1 channels) associated with microglial functions, including activation, migration, phagocytosis, and cytokine secretion [59,60,61,62,63,64,65]. Notwithstanding these findings, the individual mechanisms of each ion channel regarding microglial functions and its intracellular cascades are still poorly known.

Nevertheless, several studies revealed the mechanisms of the neurotoxic effects of activated microglia through specific ion channels and intracellular cascades. Glutamate hypersecretion from microglia in the ablation of Rhoa clearly demonstrated synaptic loss, leading to LTP impairment and memory loss, and eventually, neuronal cell death [66,67,68]. Rhoa, as one of the GTPase family regulating cytoskeleton reorganization in microglia, inhibits Src activation, which decreases Tnf production [69,70]. In a normal condition, the regulation prevents neuron from glutamate excitotoxicity triggered by Tnf elevation. Indeed, the Rhoa knockdown (Rhoa^fl/fl^:Cx3cr1^CreER+^) study demonstrated amyloidogenic processing of APP with the abundant accumulation of Aβ_1–40_ and Aβ_1–42_ peptides in the mutant brain with significant neurite damage of primary hippocampal neurons, reduction in LTP formation in the hippocampal slice, and memory deficit in behavior tests. Interestingly, pathogenic Aβ peptide accumulations conversely affect microglia activation and its detrimental progression in AD. For example, a tiny amount of application of Aβ_1–42_ oligomers in microglia showed a decrease in Rhoa activity and a following increase in Src activity. This observation could be found in APP/PS1 mice, where Rhoa activity of microglia was attenuated in the presence of abundant pathogenic Aβ peptide [68].

Some ion channels are also involved in the physiological activation of microglia. Among them, the synergistic effect of Kv1.3 and P2X4 has been well characterized [71]. Previously, purinergic receptors expressed in microglia were recognized for inflammatory response in ischemia or pain conditions. In this study, Kv1.3 regulates the resting membrane potential against the abrupt voltage change caused by calcium introduction through P2X4. This novel finding addresses the sophisticated interplay between different types of ion channels expressed in microglia for the functional stability of microglia activation. This can give an important clue for the activation mechanism of microglia [71]. Other ion channels recently discovered for their involvements in microglial activation are the hyperpolarization-activated cyclic nucleotide-gated (HCN) and Kv7/KCNQ channels. According to the calcium response to pharmacological agents, the HCN current is responsible for the inflammatory activation of microglia without phagocytic capacity, while Kv7/KCNQ is to the migratory function [72].

## 5. Paradoxical Functions of Astrocytes in Alzheimer Disease

### 5.1. Versatility of Effective Astrocyte Functions According to Environmental Changes

In response to the neuronal deterioration of AD, the feedback mechanism of cortical cells to restore the damage initiates morphological and functional changes of neurons, even reprogramming neuronal proliferation [73]. At the early stage, glia undergo significant modifications in favor of neurons, contrary to the later stage when they become overstimulated and harmful to neurons in hyperinflammatory reactions. During this period, gliosis is markedly increased with reactive astrocytes around Aβ plaques undertaking a series of phenotypic and functional changes [74]. There are, however, paradoxical functions of astrocytes depending on the neuronal environment—either beneficial or detrimental effects on neighboring neurons [3,75]. For example, it can release neuroprotective agents such as BDNF, VEGF, and bFGF, or neuroinflammatory factors including IL-1β, TNF-α, and NO [74].

Reactive astrocytes can be classified into A1 and A2 types with counteracting functions. In neuronal damage, A1 astrocytes release neurotoxins that induce rapid death of neurons and oligodendrocytes [76]. The damage of oligodendrocytes can further deteriorate neurodegeneration by spreading tau or alpha-synuclein aggregates throughout the cortical network [26,77].

Normally, the brain has a rehabilitating mechanism in response to damage like any other organs orchestrated by numerous cells and signaling molecules for the neuronal repair. In this sense, A2 astrocytes induce the release of multiple neurotrophic factors and cytokines to repair the damaged synapses in order to promote neuronal survival and regrowth. The neurotrophic factors or cytokines for that purpose encompass brain-derived neurotrophic factor (BDNF), cardiotrophin-like cytokine factor 1 (CLCF1), interleukin-6 (IL-6), and GDF15, as well as thrombospondins. Furthermore, A2 astrocytes also release neurotransmitters called gliotransmitters such as glutamate, GABA, ATP, and neuromodulators like d-serine and kynurenic acid [3]. At the same time, it prevents excitotoxicity by uptaking remnants of glutamate in the synaptic cleft [78].

### 5.2. Cellular Interactions of Reactivate Astrocytes with Neurons and Microglia through Various Cytokines, Complements, and Exosomes

In the complicated cellular environment of the cortex, various cytokines determine the functional nature of astrocytes, which also release specific cytokines to modify neurons or other glial cells. In the abundance of INF-γ, IL-1β, IL-6, and TNFα, astrocytes become activated producing ROS and NO through induction of the NFκB pathway, whereas IL-4 and IL-13 result in the opposite consequence [79]. As neurodegeneration progresses beyond the phagocytic capacity of glia, their neuroprotective effects run into the other extreme. In the activation of hyperstimulated A1 astrocyte, there is reciprocal communication between M1 microglia and A1 astrocyte. During the activation of the M1 macrophage, it releases proinflammatory factors such as TNF-α, IL-1 β, IL-6, and ROS [80]. They can provoke inflammatory cascades in the neighboring neurons as well as astrocytes. Once activated, A1 astrocytes accelerate complement cascades to cause neuronal death [76].

During the synaptic engulfment, glial cells communicate with each other through various cytokines for the most efficient synaptic rearrangement. For example, transforming growth factor β (TGF-β) released from astrocyte promotes C1q expression in neurons that is recognized by microglial complement receptor C3, thus pruning weak synapses by microglia [81]. In recent years, the effect of interleukin-33 (IL-33) from astrocytes has been well introduced for its influence on synaptic engulfment [82]. In normal development, IL-33 expressed from astrocyte contributes to the control of the number of synapses and efficient arborization when it activates nuclear factor kappa B (NF-κB) signaling inside neurons expressing its obligate receptor, IL1RL1 (ST2). It can also activate microglial polarization to the M2 type through the IL-33/ST2 signaling pathway [83]. Another cytokine from IL-1β expressed from microglia can trigger A1 stimulation through the same NF-κB signaling inducing an inflammatory amplification loop [84].

In terms of the synaptic loss by activated astrocytes, an interesting finding shows that phosphorylated protein kinase R-like endoplasmic reticulum kinase (PERK-P) activation in astrocytes leads to the unfolded protein (UPR)-reactivity state. At this stage, the active astrocytes produce different secretomes, reducing protein synthesis for synapse formation, resulting in the synaptic loss found in AD [85].

Like microglia, astrocytes also actively participate in complement signaling during the pathogenesis of AD. Previously, both neurons and glia are known to express complement proteins obscuring the understanding of signaling crosstalk. According to the study of Hong et al., astrocytes primarily express C3 under physiological conditions [86]. Aβ accumulation in the environment stimulates NF-kB in the astrocytes, resulting in C3 overproduction which activates C3aR in neurons and microglia. The C3aR activation causes a reduction in the opsonizing capacity of Aβ by microglia and cognitive defects. These adverse conditions are reversed by C3aR inhibition, suggesting a potential therapeutic development.

With the advancement of sorting technology of biological components even in nanoscale, exosomes of about nm diameter are found to involve various roles in neurogenesis as well as neurodegeneration. Most brain cells can release exosomes with their specific effects on target cells. Exosomes derived from astrocytes containing miRNAs also induce microglial activation [87]. For example, MiR-873a-5p from astrocyte exosomes initiates microglial M2 type conversion and reduces the production of pro-inflammatory factors by inhibiting the phosphorylation of ERK and the NF-kB signaling pathway [87,88].

During brain development, astrocyte-derived exosomes contribute significantly to axon development, synaptic, and dendritic pruning. For example, the transfer of miR-26a-5p into neurons through Aldolase C containing exosomes from astrocytes modulates neuronal morphogenesis and prunes excessive dendrites of developing hippocampal neurons [89,90]. In neuroinflammation, TNFα and IL-1β activate astrocyte, increasing the amount of miR-125a-5p and miR-16-5p in exosomes. When these miRNAs reach neurons, NTKR3 and its downstream effector Bcl2 are downregulated, resulting in reduction in dendritic growth, spike rates, and burst activity [91].

From the analysis of astrocyte-derived exosomes (ADE) in AD patients, it is noteworthy to recognize the higher level of complement factors like C1q, C4b, C3d, factor B, factor D, Bb, C3b, and C5b–C9 terminal complement complex (TCC) [92]. The neurotoxic effect of ADE is evident with the overlap of Aβ plaque density and C3/4 fragments staining in the post mortem human brain [93]. At the cellular level, C3b is delivered to neuronal surface membranes by ADEs initiating microglial cytotoxic attacks damaging neurons. It can also promote the generation of C5b–C9 TCC further exacerbating neuronal degeneration.

## 6. Receptor and Ion Channel Expressions in Astrocyte Contributing to Neuronal Stimulation

Metabotropic glutamate receptors (mGluRs) are highly expressed in the early developing hippocampus and cortex for neuronal development involved in synaptic plasticity and neuronal network construction [94]. As G-protein coupled receptors (GPCRs), they are categorized into three groups based on the signal transduction pathways and pharmacological profiles [95]. Group I contains mGluR1 and mGluR5, Gq-coupled receptors, resulting in the activation of phospholipase C (PLC), hydrolysis of phosphoinositides, release of calcium, and activation of protein kinase C (PKC) [96]. Group II (mGluR2, 3) and Group III (mGluR4, 6, 7, 8) are all Gi-coupled receptors, which are negatively coupled to adenylate cyclase [97] (Table 1).

In astrocytes, mostly mGluR3 and mGluR5 are expressed exhibiting not only as a calcium sensor but also a synaptic glutamate gauge [98,99]. Since mGluR3 and mGluR5 can be activated either by a surplus of synaptic glutamate or by glutamate released from the astrocyte itself, the overexpression could play a feedback pathway lowering glutamate release from the reactive astrocyte [100,101]. The activation of mGluR3 inhibits voltage-gated calcium entry into the astrocyte, at the same time activating MAPK and IP3 kinase pathways inducing neuroprotection. In the case of mGluR5, the intracellular signaling can regulate IP3-dependent calcium entry, increasing glutamate release from the astrocytes, which strengthens tripartite synapse consolidation in the hippocampus [102]. Even though the in vivo studies of mGluRs in AD are quite limited, it is interesting to note that Aβ increases expression of mGluR5 in an AD model [96]. mGluR3 and mGluR5 in A2 astrocytes can also increase synthesis and release of BDNF, which is a ubiquitous neurotrophic factor with a major role in neuronal plasticity, synapse formation, and long-term potentiation [103].

Various ion channels are also expressed in astrocytes contributing to its activation or settlement. Some potassium channels expressed in astrocyte membranes noticeably take part in the functional modulation of neurons by maintaining electrochemical gradients in the environment. In the case of Kir 4.1, it enables extracellular K^+^ homeostasis by uptaking excessive K^+^ often elevated in synaptic clefts [104]. By this way, it can maintain astrocyte resting membrane potential and cell volume, even facilitating glutamate uptake. Not only does it have the K^+^ buffering effect, but it also controls brain-derived neurotrophic factor (BDNF) expression in astrocytes [105]. BDNF is a neurotrophic factor affecting neuronal plasticity and synapse formation so its attenuation of expression by Kir 4.1 has significant consequences in astrocyte functions.

Calcium-activated large-conductance K+ channels (BK channels) in astrocytes are activated by the integrated neurons through glutamate receptor activation in the end foot of astrocytes following calcium influx [106]. BK channel activation elevates extracellular K^+^, which activates smooth muscle cells (SMCs) Kir channels, resulting in membrane hyperpolarization, decrease in calcium influx, and eventually, vasodilation. In this regard, BK channel activation by therapeutic agents could attenuate symptoms in ischemia-derived dementia or AD.

## 7. Discussion

With the advance of paradigm changes in the understanding of cause and progression of AD, it has become clear that not only Aβ, but also other environmental factors initiate as well as facilitate disease progression. In the AD neuronal environment, there are numerous floating neurotransmitters, cytokines, and signaling molecules on top of hazardous Aβ plaques and NTFs. These components interact primarily with neurons and at the same time, with glia. In the last decade, extensive research has been conducted to elucidate the interaction, intervention, and outcome of these cells affecting the survivability of neurons.

Microglia, as primary immune cells in the brain, patrol inside the cortex and hippocampus to prevent potential damages caused by both external and internal toxins. The behavior of microglia follows a similar pattern as macrophages react to inflammatory components from the initial recognition of epitopes, adhesion, and fusion to target proteins, all the way to lysogenic digestion. During the neurodevelopmental stage, neurons and microvessels are intermingled in a bundle form, then gradually separated from each other in the postnatal stage. From then on, neurons are situated in a very exclusive space well secluded from any other body system, owing to the blood–brain barrier (BBB), only nourished in cerebrospinal fluid (CSF). This meticulously controlled condition can provide a hygienic environment to brain cells immune to bacteria, acidity, inflammatory agents, and other harmful substances circulating in the other part of the body. Occasionally, however, there could be incidents of BBB breakdown and following infections like viral meningitis, multiple sclerosis (MS), and neuromyelitis. This type of endothelial cell damage in the brain is quite rare in normal circumstances, but the loosening of its tight junction and loss of elasticity often occur in old age. This kind of environmental factor is one of the main reasons for AD onset in addition to hereditary genetic defects.

No matter what the cause is, the infected or degenerated brain initiates self-rehabilitating mechanisms through the innate immune system and anti-inflammatory cascade. During this period, key immune players are recruited, presumably microglia as the most recognized. With strong phagocytic function, it tries to thwart toxic substances, including Aβ and NFT in the case of AD brain. It can not only directly phagocytose the pathologic Aβ oligomers but also encircle the protein entanglements segregating from neurons. At this point, it also releases neuroprotective cytokines and exosomes actively communicating with neurons and astrocytes, further drawing full benefit for neuronal sustainability. In addition to that, microglia also control neurotransmitter release like glutamate, preventing excitotoxicity. In the recent study introduced in this review, the glutamate release feedback mechanism has been characterized in terms of Rhoa activation. Once activated inside microglia, it can inhibit the downstream signaling cascades like Src and Tnf, eventually decreasing glutamate release.

Likewise, astrocytes also contribute to neuronal survival and protection, especially at the early stages of AD. By releasing cytokines, neurotrophic factors, and gliotransmitters, it can promote neuronal survival and regrowth. Same as microglia, it also regulates the amount of glutamate in the synaptic cleft by moderately releasing or uptaking the surplus with the help of mGluRs. The interaction with microglia is particularly recognizable in the synaptic pruning or engulfment. There are several cytokines for the interactions between them, among which IL-33 affects microglia polarization change into M2 type through the IL-33/ST2 signaling pathway. With this synergistic effect, they maintain favorable conditions for neurons with adequate amounts of cytokines and neurotransmitters for ideal synapse formations.

Unfortunately, the beneficial roles of microglia and astrocyte take a turn to the opposite extreme when neuroinflammation or immune reaction becomes chronic, similar to macrophages becoming cytotoxic in chronic inflammation. At a certain point, which is not clear yet, they become hyperstimulated, conveying pathologic Aβ oligomers or NFT into neighboring neurons packaged in APOE4 or exosomes. In addition, excitotoxicity is escalated by some ion channel activations like Kv1.3 and P2X4 in microglia, and Piezo 1 and L-type VGCC in astrocytes. Such channel activations lead to the continuous activation of glia releasing abundant glutamate into synapses, causing excitotoxicity and excessive synaptic engulfment. However, this conversion from beneficial to detrimental roles can be reversible. Even though there is a vast amount of descriptive knowledge in genetic and morphologic changes during the conversion, no clear answers have been suggested yet for the underlying mechanism. Gleaning from studies so far, a possible scenario can be raised in settling the threshold for the conversion point in the aspect of cellular lysogenic capacity. When glia reach their limit of phagocytic capacity, they still continue to maintain the phagocytosis function over neurotoxins beyond the intrinsic lysogenic capacity. Consequently, it could have gone through genetic modification, ending in the reactive form—the so-called M1 type. In addition, the undigested neurotoxins could be sputtered out in the milieu of neuronal environment, which can be uptaken by neurons leading to intracellular NFT formation, synaptic malfunction, uncontrolled release of neurotransmitters, and eventually, neuronal death.

The real facet of AD onset and progression has been gradually revealed after decades of studies not only about Aβ and neuron interactions but also about more profound understandings of neuronal environmental factors and glial interactions. With the introduction of novel technologies like single cell RNA seq, exosome purification, and high-resolution imaging, we can reach reality one step closer. The lessons learned in recent years of studies indicate that not only a simple neurotoxin like Aβ or NFT, but also numerous other molecules present in the brain and glial interactions, are involved in neurodegeneration. These novel understandings can be well applied to the development of effective therapeutic treatments.

## Figures and Tables

**Figure 1 ijms-21-06676-f001:**
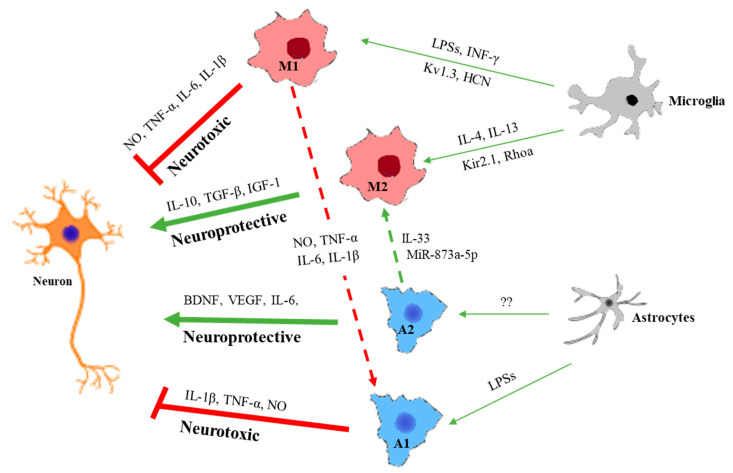
The interplay between neurons and glia in various conditions. Neuroprotective functions of activated microglia are the elimination of Aβ plaque, release of neuroprotective cytokines, and synapse pruning. However, when it turns into a neurotoxic way, it causes hyperactivation of synapse engulfment, the release of neurotoxic factors, and the involvement of tau pathology. Microglia and astrocytes can reciprocally activate each other either in the neuroprotective or neurotoxic pathway. Not only cytokines or toxins but also microRNAs can modulate the functional conversion of glia. Some ion channel expressions also contribute to the modification. NO—nitric oxide; TNF—tumor necrosis factor; IL—interleukin; LPS—lipopolysaccharides; INF—interferon; IGF—insulin growth factor; BDNF—human brain-derived neurotrophic factor; VEGF—vascular endothelial growth factor; Kv—voltage-gated potassium channels; Kir—inward-rectifier potassium channels; HCN—hyperpolarization-activated cyclic nucleotide-gated channels. NO—nitric oxide; TNF—tumor necrosis factor; IL—interleukin; LPS—lipopolysaccharides; INF—interferon; IGF—insulin growth factor; BDNF—human brain-derived neurotrophic factor; VEGF—vascular endothelial growth factor; Kv—voltage-gated potassium channels; Kir—inward-rectifier potassium channels; HCN—hyperpolarization-activated cyclic nucleotide-gated channels.

**Table 1 ijms-21-06676-t001:** Group of metabotropic glutamate receptors expressed in glia.

Groups	Subtypes	G-protein	Intracellular Signaling	Function	Location
Group I	mGluR1	Gq	PLC↑ PKC↑ Ca release ↑ Adenylate cyclase ↑	LTP facilatation	Postsynaptic forebrain/midbrain
	mGluR5				Astrocyte, postsynaptic forebrain, midbrain
Group II	mGluR2	Gi	Adenylate cyclase↓	LTD facilitation LTP inhibition	Pre/postsynaptic forebrain
	mGluR3				Astrocyte, postsynaptic forebrain
Group III	mGluR4	Gi	Adenylate cyclase↓ cGMP-PDE↓	LTD facilitation LTP inhibition	Pre/postsynaptic cerebellum
	mGluR6				Postsynaptic, retina
	mGluR7				Pre/postsynaptic
	mGluR8				Pre/postsynaptic, spinal cord

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
