# Peer review of "Deleterious Alteration of Glia in the Brain of Alzheimer’s Disease"

_ijms, 2020, doi:10.3390/ijms21186676_

Round 1

Author Response

This review documents that not only Aβ, but additional factors are entirely involved in the progression of the Alzheimer disease; there are numerous floating neurotransmitters, cytokines, signalling molecules on top of hazardous Aβ plaques and NTFs. The description of these issues, however, overwhelmed with details in many cases, thus they become difficulty their understanding. A typical example is in the text line 2015-226.

We deleted ‘transgenic APPswe/PS1dE9’ in the sentence below to remove the detail of information, which can be found in the reference.

 Even though the in vivo studies of mGluRs in AD are quite limited, it is interesting to note that Aβ increases expression of mGluR5 in an AD transgenic APPswe/PS1dE9 model.

  • Line 286: Even though the in vivo studies of mGluRs in AD are quite limited, it is interesting to note that Aβ increases expression of mGluR5 in an AD model.

It is clear, that „the real facet of AD onset and progression still remain enigmatic even after decades of studies from almost every aspect of basic and clinical research”, nevertheless, there are published data referring to the effects of molecular level.

We modified the sentence as below:

The real facet of AD onset and progression still remain enigmatic even after decades of studies from almost every aspect of basic and clinical research.

  • Line 367-369: The real facet of AD onset and progression has been gradually revealed still remain enigmatic even after decades of studies not only from Aβ and neuron interactions but also from more profound understandings of neuronal environmental factors and glial interactions.

For example, as described in 244-248 lines „A recently published study showed synaptic loss by activated astrocyte……. indicates that the intrinsic, as well as intercellular modification of astrocytes, can degenerate neurons into severe stage eventually to AD.” However, Smith et al. (Neuron 2020) reported that astrocyte unfolded protein can induce a specific reactivity causing neuronal degeneration resulting from UPR over-activation, a distinct pathogenic mechanism the targeted of which can be effective for neuroprotection.

We modified the paragraph as below:

A recently published study showed synaptic loss by activated astrocyte [86]. In this study, phosphorylated protein kinase R-like endoplasmic reticulum (ER) kinase (PERK-P) phosphorylates eukaryotic initiation factor 2 (eIF2α) reducing protein synthesis rates involved in synapse synthesis resulting in synaptic loss found in AD [86]. This indicates that the intrinsic as well as intercellular modification of astrocytes can degenerate neurons into severe stage eventually to AD.

  • Line 235-239: In terms of the synaptic loss by activated astrocyte, an interesting finding shows that phosphorylated protein kinase R-like endoplasmic reticulum kinase (PERK-P) activation in astrocytes leads to the unfolded protein (UPR)-reactivity state. At this stage, the active astrocytes produce different secretomes, reducing protein synthesis for synapse formation resulting in synaptic loss found in AD [84].

In addition, there are poorly documented, if at all, the interrelationship of the amyloid aggregates and the environment factors leading to the aetiology of the AD. Is that unambiguous what is (are) the initiator(s) of the progression? It is a critic issue concerning the evaluation of the therapeutic strategy.

We added a paragraph like below:  

Line 243-248: Aβ accumulation in the environment stimulates NF-kB in the astrocytes resulting in C3 overproduction which activates C3aR in neurons and microglia. The C3aR activation causes reduction of opsonizing capacity of Aβ by microglia and cognitive defects. These adverse conditions are reversed by C3aR inhibition suggesting a potential therapeutic development. 

In addition, there are poorly, if at all, description about the potential interrelationship of oligodendrocytes in the progression of AD to their communication with the neurons as reported in relation to the progression of synucleinopathies.

We modified the sentence as below:

Line 202: The damage of oligodendrocytes can further deteriorate neurodegeneration by spreading tau or alpha-synuclein aggregates throughout the cortical network

Reviewer 2 Report

The review written by Kim et al., aims at providing a concise overview of the latest progress in the study of neuroinflammatory interaction in AD progression.

Overall, the manuscript has some good insights but lacks a consistent flow and a few major points need to be answered or re-written.

Major points:

• Page 3 Line 124: “In the most recent studies, two different groups suggest contradictory

results about the interaction of tau oligomers and microglia”

This sentence is misleading as the two studies the authors are referring to BOTH lead to

the conclusion that microglia are not the instigators of AD progression and instead (at

least in the initial phase of disease progression) are activated only when pTau is present.

• Page 3 Lines 118-139: The earlier paragraph focused more on the complement systemmicroglia-

Ab interaction. So, in keeping with the flow, it would be better if the authors

added information on hand regarding the effects of the complement system and

microglia on pTau (example: cx3cl1).

• The content under “Microglia polarization and its detrimental effect in neuronal

homeostasis during AD progression” mainly talks about the complement pathway.

However, this pathway alone is not responsible for glial activation in AD. Cellular as well

as inflammasome pathways have been documented in the context of Alzheimer’s disease

and microglia activation.

Either the authors should add information regarding the other pathways or change the

title to better reflect their original content.

• Page 4 line 181: II. Paradoxical function of astrocytes in the brain of Alzheimer disease.

Again, there is not much flow between the previous paragraph (regarding microgliacomplement/

cytokine and ion channel expression in early and late AD) and this paragraph

on astrocytes. The authors barely mention the cytokines involved in AD pathogenesis and

there is no mention of ion channels up/downregulated during disease progression (please

look into inward rectifying potassium channel, Kir4.1 and the BK calcium-dependent

potassium channel).

• Page 6, line 258: Not sure what this sentence means, “Aldolase C (AldoC), a

glycolytic/gluconeogenic enzyme primarily expressed in forebrain astrocytes are

packaged into exosome modulating miR-26a-5p content during the development of

primary cortical neurons” Does AldoC that is packaged in exosomes, modulate the miRNA

miR-26a-5p? If so, where is this miRNA expressed? What exactly does this “modulation”

do to the developing neurons? Please re-write this sentence.

• Page 6, line 235: “For example, transforming growth factor β (TGF-β) released from

astrocyte promotes phagocytosis of microglia by regulating C1q expression” Are the

microglia phagocytosed by astrocytes? Or do the authors mean to say that there is a direct

relation between TGF-β and C1q expression and higher phagocytosis BY microglia?

• Page 6, line 249: Again in keeping with previous content- there should be some mention

of complement factors secreted from astrocytes through exosomes (see PMID:

26758846, 29406582)

Minor points

• Please check the whole document for grammar and sentence structure. Some paragraphs

have markedly more sentence structure errors than others.

Author Response

The review written by Kim et al., aims at providing a concise overview of the latest progress in the study of neuroinflammatory interaction in AD progression.   

Overall, the manuscript has some good insights but lacks a consistent flow and a few major points need to be answered or re-written.

Major points:

  • Page 3 Line 124: “In the most recent studies, two different groups suggest contradictory results about the interaction of tau oligomers and microglia”

This sentence is misleading as the two studies the authors are referring to BOTH lead to the conclusion that microglia are not the instigators of AD progression and instead (at least in the initial phase of disease progression) are activated only when pTau is present.

We changed the sentence as below:

Some studies showed that tau pathology also contributes to activate the complement system suggesting exacerbation of tau pathology.

  • Page 3 Lines 118-139: The earlier paragraph focused more on the complement systemmicroglia-Ab So, in keeping with the flow, it would be better if the authors added information on hand regarding the effects of the complement system and microglia on pTau (example: cx3cl1).

We added new paragraph as below:

Line 126-130: The role of the CX3CR1 signaling pathway of microglia has been demonstrated as a key factor to phagocytose tau entanglements [52]. In normal neuronal development, cytokine CX3CL1 (fractalkine) expressed by neurons binds to CX3CR1 of microglia maintaining resting state, leading to inhibiting pro-inflammatory cytokine release [53]. In AD condition, on the other hand, tau directly binds to CX3CR1 breaking the resting state of microglia, which is related to tau internalization by microglia [52].

  • The content under “Microglia polarization and its detrimental effect in neuronal homeostasis during AD progression” mainly talks about the complement pathway. However, this pathway alone is not responsible for glial activation in AD. Cellular as well as inflammasome pathways have been documented in the context of Alzheimer’s disease and microglia activation.

Either the authors should add information regarding the other pathways or change the title to better reflect their original content.

We changed the title as below:

  1. Complement pathway of microglia in the detrimental effect during AD progression

  • Page 4 line 181: II. Paradoxical function of astrocytes in the brain of Alzheimer disease. Again, there is not much flow between the previous paragraph (regarding microgliacomplement/cytokine and ion channel expression in early and late AD) and this paragraph on astrocytes. The authors barely mention the cytokines involved in AD pathogenesis and there is no mention of ion channels up/downregulated during disease progression (please look into inward rectifying potassium channel, Kir4.1 and the BK calcium-dependent potassium channel).

We reorganized the order of contents parallel to the microglia section.

  1. Paradoxical function of astrocytes in the brain of Alzheimer disease
  2. Versatility of effective astrocyte functions according to environmental changes
  3. Cellular interactions of reactivate astrocytes with neurons and microglia through various cytokines, complements, and exosomes
  4. Receptor and ion channel expressions in astrocyte contributing to neuronal stimulation

Cytokines and ion channel stories are included under the designate titles.

  • Page 6, line 258: Not sure what this sentence means, “Aldolase C (AldoC), a glycolytic/gluconeogenic enzyme primarily expressed in forebrain astrocytes are packaged into exosome modulating miR-26a-5p content during the development of primary cortical neurons” Does AldoC that is packaged in exosomes, modulate the miRNA miR-26a-5p? If so, where is this miRNA expressed? What exactly does this “modulation” do to the developing neurons? Please re-write this sentence.

We changed the sentence as below:

For example, the transfer of miR-26a-5p into neurons through Aldolase C containing exosomes from astrocytes modulates neuronal morphogenesis and prunes excessive dendrites of developing hippocampal neurons [98,99].

  • Page 6, line 235: “For example, transforming growth factor β (TGF-β) released from astrocyte promotes phagocytosis of microglia by regulating C1q expression” Are the microglia phagocytosed by astrocytes? Or do the authors mean to say that there is a direct relation between TGF-β and C1q expression and higher phagocytosis BY microglia?

We changed the sentence as below:

For example, transforming growth factor β (TGF-β) released from astrocyte promotes C1q expression in neurons that is recognized by microglial complement receptor C3, thus pruning weak synapses by microglia [80].

  • Page 6, line 249: Again in keeping with previous content- there should be some mention of complement factors secreted from astrocytes through exosomes (see PMID:

26758846, 29406582) 

We added new paragraph as below:

PMID: 26758846

Line 240: Like microglia, astrocytes also actively participate in complement signaling during the pathogenesis of AD. Previously, both neurons and glia are known to express complement proteins obscuring the understanding of signaling cross-talk. According to the study of Hong et al., astrocytes primarily express C3 under physiological conditions [85].

PMID: 29406582

Line 261: From the analysis of astrocyte-derived exosome (ADE) in AD patients, it is noteworthy to recognize the higher level of complement factors like C1q, C4b, C3d, factor B, factor D, Bb, C3b, and C5b-C9 terminal complement complex (TCC) [86].

Minor points   

  • Please check the whole document for grammar and sentence structure. Some paragraphs have markedly more sentence structure errors than others.

            We have gone through all the grammatical errors and fixed sentences.

Round 2

Reviewer 1 Report

accepted

Author Response

We greatly appreciate to your effort.

Reviewer 2 Report

Please make the following changes

  • Some studies showed that tau pathology also contributes to activate the complement system suggesting exacerbation of tau pathology. 

        This sentence does not make sense. Please do not simply change just one sentence- you should briefly mention (with appropriate citations)  what happens/ how the tau aggregation/phosphorylation leads to complement system activation and subsequent neuroinflammation or protein aggregation or cell death.

The authors made the appropriate changes to other parts. Aside from the point mentioned above, no further changes are necessary.

Author Response

We added new paragraph in line 121-130.

Interestingly, many evidences address the tight correlation between the complement pathway and tau pathology. In the human AD brain and tauopathy mouse model, C1q levels were positively correlated with the tau aggregation, and C3-C3aR1 expressions were increased proportional to the aggravation of tau pathology harming the cognitive function [45,46]. Since microglia likely phagocytose C1q-tagged synapses leading to synapse loss, tau entanglements induce accumulation of C1q at the synapse [45]. Once the complement cascade is initiated by the activation of C1q, C3-C3aR1 signaling can exacerbate tau pathology through the signal transducer and activator of transcription 3 (STAT3) pathway inducing neuroinflammation [46]. Thus, inhibition of the complement pathway can be suggested for reduction of tauopathy.